# Rapid changes in tissue mechanics regulate cell behaviour in the developing embryonic brain

**Amelia J Thompson[†], Eva K Pillai[†], Ivan B Dimov, Sarah K Foster, Christine E Holt, Kristian Franze***

Department of Physiology, Development and Neuroscience, University of Cambridge, Cambridge, United Kingdom

**Abstract** Tissue mechanics is important for development; however, the spatio-temporal dynamics of in vivo tissue stiffness is still poorly understood. We here developed tiv-AFM, combining time-lapse in vivo atomic force microscopy with upright fluorescence imaging of embryonic tissue, to show that during development local tissue stiffness changes significantly within tens of minutes. Within this time frame, a stiffness gradient arose in the developing *Xenopus* brain, and retinal ganglion cell axons turned to follow this gradient. Changes in local tissue stiffness were largely governed by cell proliferation, as perturbation of mitosis diminished both the stiffness gradient and the caudal turn of axons found in control brains. Hence, we identified a close relationship between the dynamics of tissue mechanics and developmental processes, underpinning the importance of time-resolved stiffness measurements.

DOI: https://doi.org/10.7554/eLife.39356.001

***For correspondence:**
kf284@cam.ac.uk

[†]These authors contributed equally to this work

**Competing interests:** The authors declare that no competing interests exist.

During embryonic development, many biological processes are regulated by tissue mechanics, including cell migration (*Barriga et al., 2018*), neuronal growth (*Koser et al., 2016*), and large-scale tissue remodelling (*Butler et al., 2009*; *Munjal et al., 2015*). Recent measurements at specific time points suggested that tissue mechanics change during developmental (*Koser et al., 2016*; *Iwashita et al., 2014*; *Majkut et al., 2013*) and pathological (*Murphy et al., 2011*; *Moeendarbary et al., 2017*) processes, which might significantly impact cell function. Furthermore, several approaches have recently been developed to measure in vivo tissue stiffness, including atomic force microscopy (*Barriga et al., 2018*; *Koser et al., 2016*; *Gautier et al., 2015*), magnetic resonance elastography (*Sack et al., 2008*), Brillouin microscopy (*Scarcelli and Yun, 2012*), and magnetically responsive ferrofluid microdroplets (*Serwane et al., 2017*). However, the precise spatiotemporal dynamics of tissue mechanics remains poorly understood, and how cells respond to changes in local tissue stiffness in vivo is largely unknown.

To enable time-resolved measurements of developmental tissue mechanics, we here developed time-lapse in vivo atomic force microscopy (tiv-AFM), a method that combines sensitive upright epi-fluorescence imaging of opaque samples, such as frog embryos, with iterated AFM indentation measurements of in vivo tissue at cellular resolution and at a time scale of tens of minutes (*Figure 1*). A fluorescence zoom stereomicroscope equipped with an sCMOS camera (quantum yield 82%) was custom-fitted above a bio-AFM set-up (*Figure 1—figure supplement 1*), which had a transparent pathway along the area of the cantilever. To cope with the long working distance required for imaging through the AFM head, the microscope was fitted with a 0.125 NA/114 mm WD objective. The AFM was set up on an automated motorised stage containing a temperature-controlled sample holder to maintain live specimens at optimal conditions during the experimental time course. (*Figure 1a,b*) (see Materials and methods for details).

**eLife digest** Neurons in the brain form an intricate network that follows a precise template. For example, in a young frog embryo, the neurons from the eyes send out thin structures, called axons, which navigate along a well-defined path and eventually connect with the visual centres of the brain. This journey requires the axons to take a sharp turn so they can wire with the right brain structures.

Axons find their paths not only by following chemical signals but also by reacting to the stiffness of their environment. In an older frog embryo for instance, the brain is stiffer at the front, and softer at the back. As neurons from the eyes make their way through the brain, they turn to follow this gradient, moving away from stiffer areas towards the softer regions.

Here, Thompson, Pillai et al. investigate when and how this stiffness gradient is established in frogs. To do so, a new technique was developed. Called time-lapse in vivo atomic force microscopy, the method measures how brain stiffness changes over time in a live embryo, while also taking images of the growing axons.

The experiments show that the stiffness gradient arose within tens of minutes, just as the first 'pioneering' axons from the eyes began to grow across the brain. These axons then responded to the gradient, turning towards the softer tissue. Changes in the number of cells in the underlying brain tissue governed the formation of the gradient, with rapidly stiffening areas containing more cells than those that remained soft. In fact, using drugs that stop cells from dividing reduced both the mechanical gradient and the turning response of the axons.

The technique developed by Thompson, Pillai et al. is a useful tool that can help elucidate how variations in stiffness control the brain wiring process. It could also be used to look into how other developmental or regenerative processes, such as the way neurons reconnect after injuries to the brain or spinal cord, may be regulated by mechanical tissue properties.

DOI: https://doi.org/10.7554/eLife.39356.002

We tested tiv-AFM using the developing *Xenopus* embryo brain during outgrowth of the optic tract (OT) as a model (*Figure 1c*). In the OT, retinal ganglion cell (RGC) axons grow in a bundle across the brain surface, making a stereotypical turn in the caudal direction *en route* that directs them to their target, the optic tectum (*McFarlane and Lom, 2012*). We previously demonstrated that by later stages of OT outgrowth (i.e. when axons had reached their target), a local stiffness gradient lies orthogonal to the path of OT axons, with the stiffer region rostral to the OT and softer region caudal to it (*Koser et al., 2016*). This gradient strongly correlated with axon turning, with the OT routinely turning caudally towards softer tissue (*Koser et al., 2016*). We therefore wanted to determine when this stiffness gradient first developed, whether its emergence preceded OT axon turning, and what the origin of the stiffness gradient was.

To answer these questions, we performed iterated tiv-AFM measurements of the embryonic brain in vivo at early-intermediate stages, that is, just before and during turn initiation by the first 'pioneer' OT axons. The apparent elastic modulus $K$, which is a measure of the tissue's elastic stiffness, was assessed in an ~150 by 250 μm raster at 20 μm resolution every ~35 min, producing a sequence of 'stiffness maps' of the area (*Figure 2—figure supplement 1*). The applied force ($F$ = 10 nN) and cantilever probe ($r$ = 18.64 μm) were chosen to measure the stiffness mainly of the top ~20–30 μm of the tissue, within which RGC axons grow (*Holt, 1989*; *Harris et al., 1987*). To reduce noise, raw AFM data were interpolated and smoothed in $x$-, $y$-, and time dimensions using an algorithm based on the discrete cosine transform (*Figure 2a,b*, see Materials and methods for details) (*Garcia, 2010*; *Garcia, 2011*). Simultaneously, we recorded optical time-lapse images of fluorescently labelled RGC axons growing through the region of interest (*Figure 2a*, *Figure 2—figure supplement 1*).

To assess whether repeated AFM measurements of the *Xenopus* brain affect RGC axon growth, we first conducted time-lapse AFM measurements on one group of embryos while we exposed another group to the same conditions without making force measurements. At the end of the experiments (i.e., at stage 37/38), OTs were labelled with DiI (*Wizenmann et al., 2009*), and their elongation and turning angles measured. We did not find any significant differences between the groups (*Figure 2—figure supplement 2*), suggesting that repeated AFM measurements do not affect axon growth.

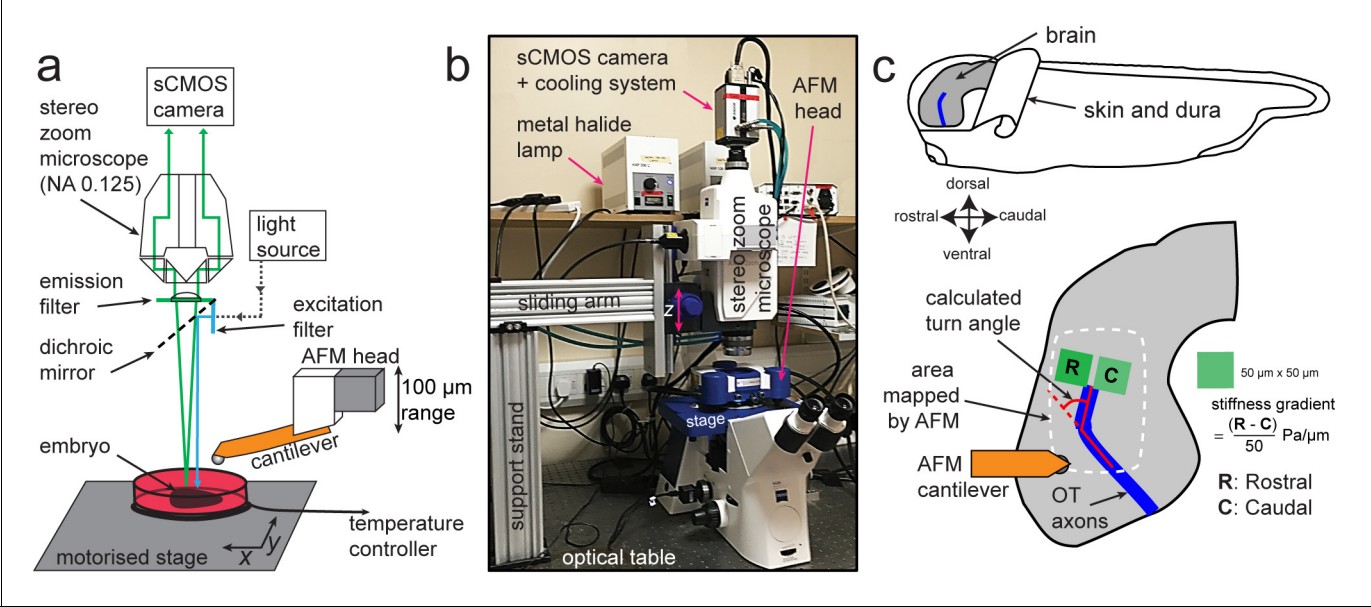

**Figure 1.** Experimental set-up for combined time-lapse in vivo AFM (tiv-AFM). (**a**) Schematic (not to scale) and (**b**) photograph of the experimental setup. An AFM with 100 µm z-piezo range is positioned above a temperature-controlled sample chamber containing the specimen. A custom-fit fluorescence zoom stereomicroscope with a long (114 mm) working distance and NA 0.125 objective, connected to a high quantum-efficiency sCMOS camera, is mounted on a custom-built support stand above the AFM head optimised for trans-illumination. The specimen is moved by a motorised x/y stage to allow AFM-based mapping of large areas. (**c**) (Top) Schematic of a *Xenopus* embryo, showing both how the brain is prepared for tiv-AFM and rostral-caudal (R/C) and dorsal-ventral (D/V) embryonic axes. All following images of embryonic brains in vivo will have the same orientation. (Bottom) Close-up diagram of the brain, showing the approximate region mapped by AFM (white dashed line), within which optic tract (OT) axons (blue) turn caudally. Also shown are the regions of interest (green boxes) used to calculate brain stiffness rostral and caudal of the OT, and hence the developing stiffness gradient. Red overlaid lines show calculation of the angle through which OT axons turn (turn angle).
DOI: https://doi.org/10.7554/eLife.39356.003

The following figure supplement is available for figure 1:

**Figure supplement 1.** Custom-built support stand for the upright optical imaging set-up.
DOI: https://doi.org/10.7554/eLife.39356.004

We then performed tiv-AFM measurements of developing *Xenopus* brains. Early in the time-lapse sequence (i.e. prior to axon turning), the stiffness of the brain was similar on both sides of the OT. However, over the time course of the measurements a stiffness gradient arose, mostly due to rising stiffness of tissue rostral to the OT (*Figure 2a,b*). Visual inspection of the fold-change in tissue stiffness from one time point to the next indicated that significant changes in tissue mechanics were already occurring approximately 40–80 min after the onset of measurements (*Figure 2b*), that is before axons started turning caudally, suggesting that the tissue stiffness gradient was established prior to axon turning.

To test this hypothesis, we quantified the temporal evolution of the stiffness gradient in a small region immediately in front of the advancing OT (*Figure 2—figure supplement 1a*). At the beginning of each time point in the sequence of tiv-AFM maps, we calculated the angle through which axons turned ('OT turn angle'). For each animal, minimum and maximum absolute values were rescaled to 0 and 1, respectively (*Figure 2c*). The projected appearance of the stiffness gradient preceded the projected onset of axon turning on average by 18 min (*Figure 2d,e*), indicating that axons indeed turned after the stiffness gradient was established, which is consistent with a role for mechanical gradients in helping to guide OT axons caudally (*Koser et al., 2016*). Based on the first time point at which we detected axon turning in each animal, our data suggested that a stiffness gradient of at least (0.9 ± 0.4) Pa/µm (mean ± SEM) was required for axons to change their growth direction. In line with this idea, RGC axons from heterochronic eye primordia transplants growing through *Xenopus* brains at stages before the stiffness gradient is established grow rather straight and do not turn caudally in the mid-diencephalon (*Cornel and Holt, 1992*).

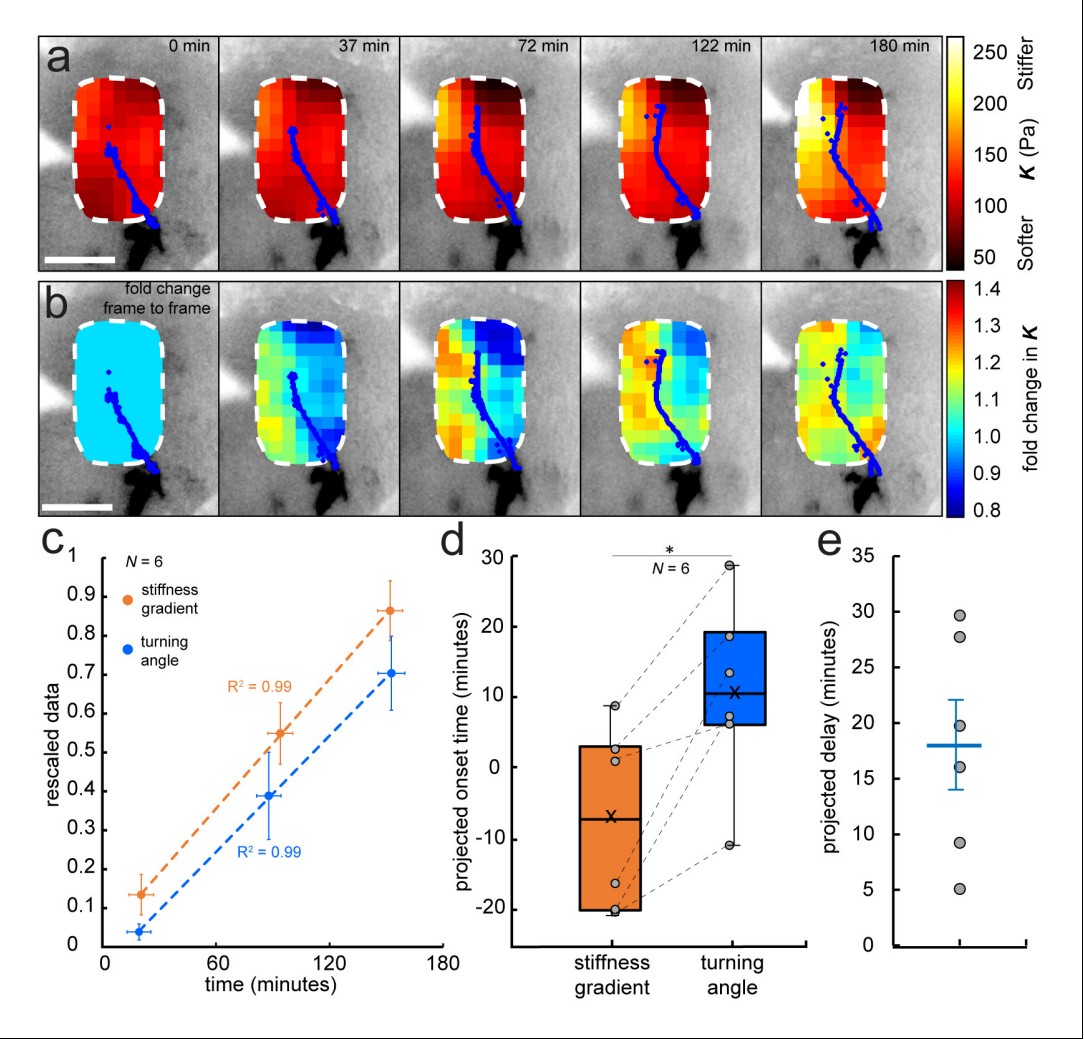

**Figure 2.** Development of a stiffness gradient in the *Xenopus* embryo brain precedes axon turning. (a) Time-lapse stiffness maps obtained from a tiv-AFM experiment, showing outlines of fluorescently labelled OT axons (blue) and processed AFM-based stiffness maps (colour maps) overlaid on images of the brain. Colour maps encode the apparent elastic modulus $K$, a measure of tissue stiffness, assessed at an indentation force $F = 10$ nN. The time in minutes on each frame is taken from the timestamp of the first measurement in each successive stiffness map; the corresponding overlaid fluorescence images were obtained simultaneously. (b) Visualisation of fold-changes in brain tissue stiffness from one time point to the next, based on the interpolated and smoothed data shown in *Figure 2a*. Colour scale encodes the fold-change in $K$ at each location on the stiffness map, expressed relative to the values at the previous time point, with the exception of t = 0 min, where all values were set to 1.Tissue stiffness changes throughout the time course, with large changes already occurring between ~40–80 min after the start of the experiment. (c) Plot of mean re-scaled values for the stiffness gradient (orange) and OT turn angle (blue). Stiffness values were binned to match the time points of the developmental stages at which cell body densities were assessed. Dashed lines denote linear fits ($R^2 = 0.99$). (d) Boxplots of the extrapolated appearance times of the stiffness gradients and the onset of OT axon turning, relative to the start time of tiv-AFM measurements, with ladder plots for individual embryos overlaid (grey circles/dashed lines). Extrapolations are based on linear fits to the re-scaled data for individual animals (*Figure 2c*). Stiffness gradients appear significantly earlier than the onset of axon turning (p=0.03, paired Wilcoxon signed-rank test). (e) Scatterplot showing the time delay between extrapolated onsets of stiffness gradients and axon turning, calculated for individual animals. The average delay of 18 min is indicated by the blue line. Boxplots show median, first, and third quartiles; whiskers show the spread of the data; '×' indicates the mean. Error bars denote standard error of the mean. *p<0.05. AFM measurement resolution, 20 µm; all scale bars, 100 µm. $N$ denotes number of animals.

DOI: https://doi.org/10.7554/eLife.39356.005

The following figure supplements are available for figure 2:

*Figure 2 continued on next page*

*Figure 2 continued*

**Figure supplement 1.** Data processing for tiv-AFM experiments.
DOI: https://doi.org/10.7554/eLife.39356.006
**Figure supplement 2.** Iterated AFM measurements of brain in vivo do not affect OT outgrowth.
DOI: https://doi.org/10.7554/eLife.39356.007
**Figure supplement 3.** Stereotypy of brain stiffness changes relative to OT development.
DOI: https://doi.org/10.7554/eLife.39356.008

We have previously shown that tissue stiffness scales with local cell body density (*Barriga et al., 2018*; *Koser et al., 2015*), and that in *Xenopus* embryo brains local stiffness gradients at later developmental stages (39-40) correlate with a gradient in cell density (*Koser et al., 2016*). To determine if changes in cell densities are driving changes in tissue stiffness, and thus parallel the evolution of the stiffness gradient at earlier stages, we assayed cell densities using DAPI labelling of nuclei in whole-mounted brains with fluorescently labelled OTs, beginning at the morphological stage corresponding to the start of tiv-AFM measurements (33/34) and repeated for the two subsequent stages encompassing OT turning (35/36 and 37/38).

While at the first stage cell densities on both sides of the OT were similar, a clear difference in nuclear densities rostral and caudal to the OT developed at later stages (*Figure 3a*). Cell densities at the two later stages were significantly higher in the region rostral to the OT (i.e. where tissue was stiffer) than caudal to it, and the overall magnitude of the cell density gradient significantly rose over time (*Figure 3b*). Plotting the stage-specific gradient in cell body densities against the stiffness gradient revealed a strong linear correlation between them (Pearson's correlation coefficient ρ=0.97) (*Figure 3c*).

To test if local cell densities drive the evolution of the stiffness gradient during OT turning, we repeated both nuclear staining and tiv-AFM measurements on embryos treated with the mitotic blocker BI2536 (*Lénárt et al., 2007*), which inhibits Polo-like kinase 1 and has previously been used to inhibit in vivo cell proliferation in the embryonic retina (*Weber et al., 2014*). BI2536 also triggered mitotic arrest in brains of developing *Xenopus* embryos, as the number of phosphorylated histone H3 (pH3)-positive cells (*Hugle et al., 2015*) was significantly higher in treated compared to control brains, and the cross-sectional area of treated brains was significantly decreased at later stages, indicating a decrease in total cell number (*Figure 4a–c*). Inspection of the stage-dependent distribution of cells in the developing *Xenopus* brain suggested that in BI2536-treated brains, the nuclear density was decreased particularly rostral to the OT (*Figure 4d*, *Figure 4—figure supplement 1*). In tiv-AFM experiments, blocking cell proliferation significantly attenuated the increase of both the stiffness gradient and the OT turn angle over the time course of the experiment (*Figure 4e,f*), suggesting that the gradient in cell densities strongly contributed to the stiffness gradient, which in turn helps instruct axon growth.

The absence of an increase in tissue stiffness near the advancing OT in BI2536-treated brains was furthermore accompanied by a decrease in OT elongation (*Figure 4g*). Similarly, OT elongation decreased when brains were softened by manipulating the extracellular matrix (*Koser et al., 2016*), confirming that tissue stiffness is involved in regulating axon growth in vivo.

We obtained similar results using a different mitotic blocker, hydroxyurea/aphidicolin (HUA), which inhibits DNA replication (*Gilman et al., 1980*; *Ikegami et al., 1978*) and has previously been used to block cell division in *Xenopus* embryos (*Harris and Hartenstein, 1991*). HUA also decreased the gradient in nuclear densities (mainly by reducing nuclear densities rostral to the OT), decreased brain stiffness, and generated defects in OT outgrowth (*Figure 4—figure supplement 2*).

In order to test if the mitotic blocker BI2536 perturbs neuronal mechanosensing, we cultured eye primordia on laminin-coated polyacrylamide substrates of different stiffnesses (*Koser et al., 2016*), exposed the outgrowing RGC axons to the drug, and quantified axon growth as a function of substrate stiffness. While RGC axon growth on stiff substrates with a shear modulus $G'{\sim}5{,}500$ Pa was not altered by the presence of BI2536 if compared to control conditions (*Figure 4—figure supplement 3e*), axons grew longer on soft substrates of $G'{\sim}200$ Pa when they were exposed to 50 nM BI2536 but not when they were exposed to 500 nM (*Figure 4—figure supplement 3f*), suggesting that the drug might partially impact axon growth on very soft substrates (although no significant differences were observed when axons were grown on slightly stiffer substrates of $G'{\sim}300$ Pa,

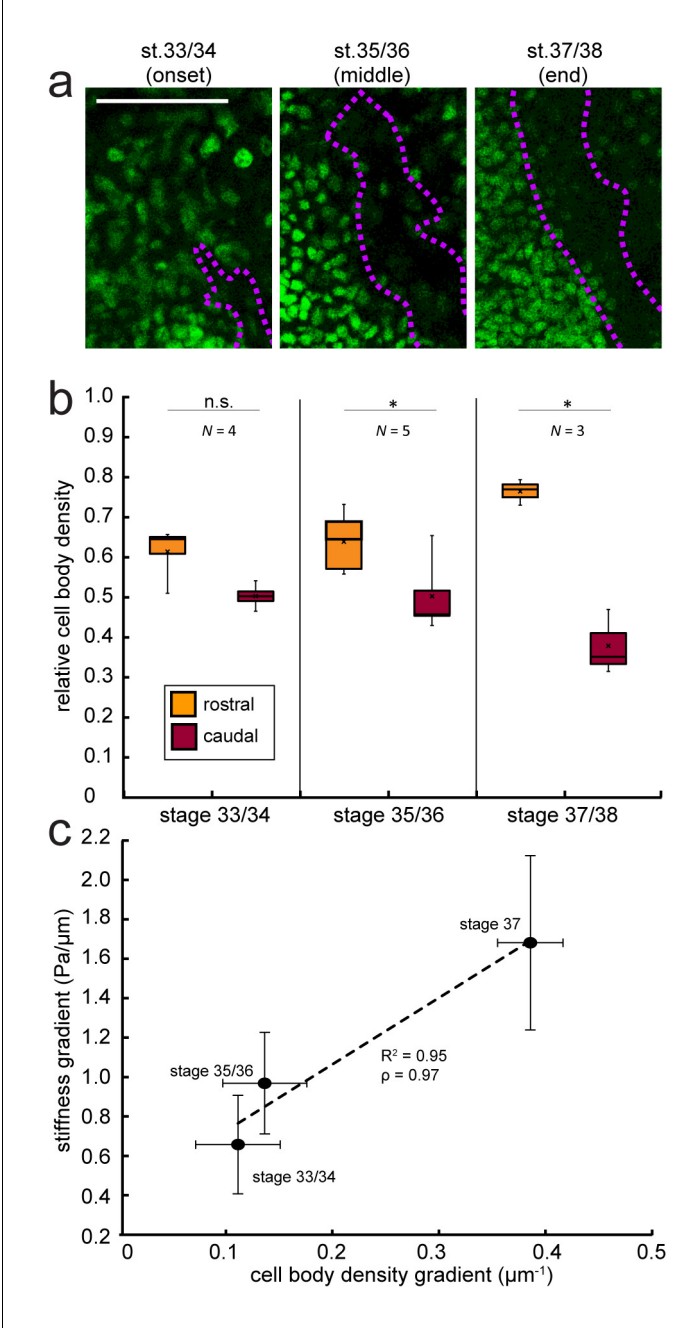

**Figure 3.** Changes in local cell body densities contribute to the emerging in vivo stiffness gradient in the *Xenopus* embryo brain. (a) Immunohistochemistry of nuclei (green) in whole-mount control *Xenopus* embryo brains at successive developmental stages. Stages shown correspond to the onset (stage 33/34), approximate middle (stage 35/36), and end (stage 37/38) of tiv-AFM measurements. OT axons are outlined in purple. (b) Local cell body densities were significantly higher rostral to the OT than caudal to it at both stage 35/36 (p=0.03, paired Wilcoxon signed-rank test) and stage 37/38 (p=0.04). (c) Gradients in local cell body density and tissue stiffness strongly correlate with each other (Pearson's correlation coefficient ρ = 0.97). Binned absolute values for the stiffness gradient (in Pa/μm) are plotted against the mean cell density gradient at each developmental stage. Dashed line denotes linear fit ($R^2$ = 0.95). Boxplots show median, first, and third quartiles; whiskers show the spread of the data; '×' indicates the mean. Error bars denote standard error of the mean. *p<0.05. AFM measurement resolution, 20 μm; all scale bars, 100 μm. *N* denotes number of animals.

DOI: https://doi.org/10.7554/eLife.39356.009

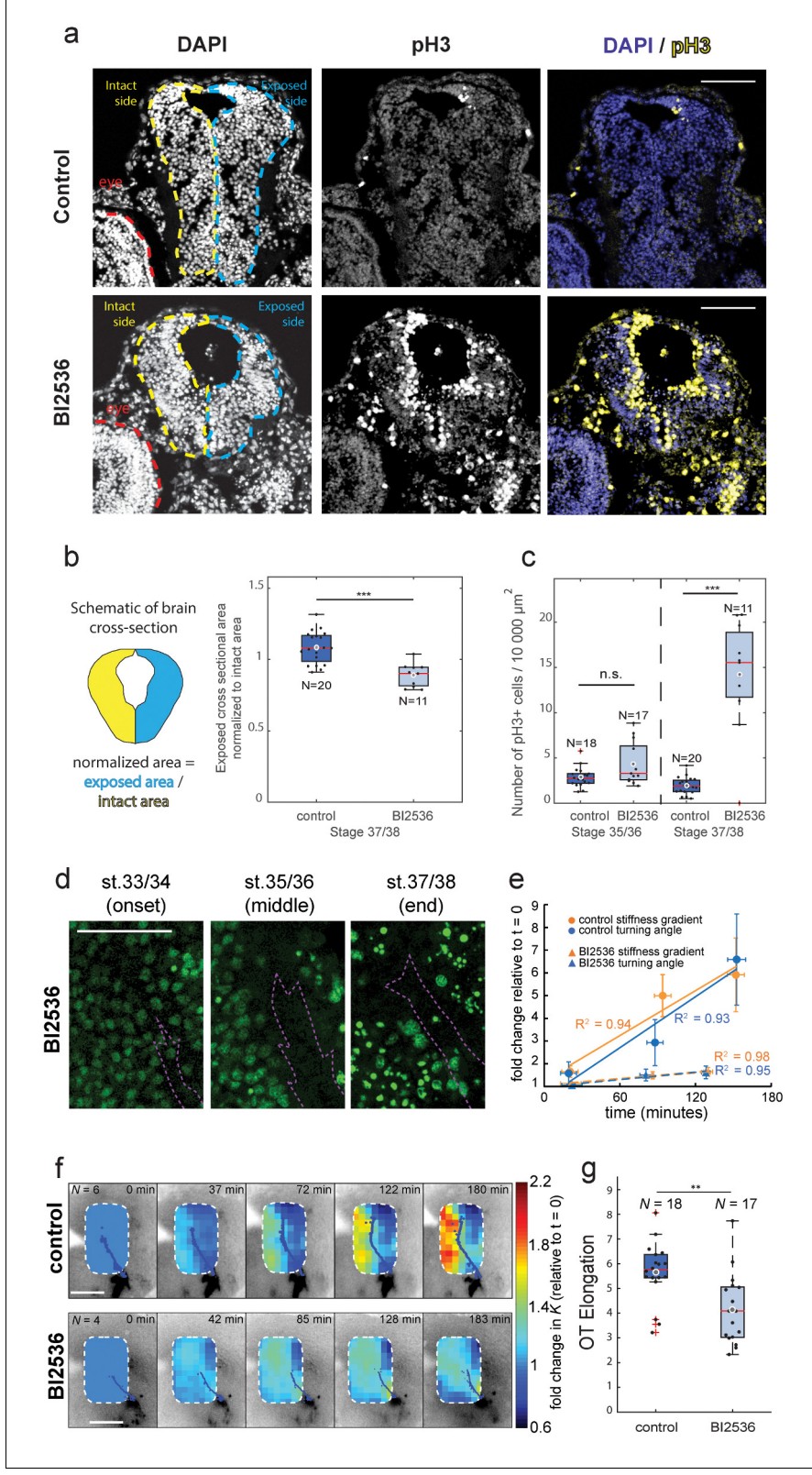

**Figure 4.** Blocking mitosis in vivo reduces local cell density, decreases mechanical gradients, and attenuates both RGC axon turning and overall OT elongation. (a) Coronal sections of stage 37/38 control and BI2536-treated embryos stained for DAPI and phospho-histone3 (pH3). (b) Boxplot of normalized exposed brain area. The schematic demonstrates how regions were normalized. Normalized brain area is significantly lower in the BI2536-

*Figure 4 continued*

treated embryos compared with controls (p=5.7e-05, Wilcoxon rank-sum test), indicating a decrease in total cell number. (c) The number of pH3+ cells per 10 000 $\mu m^2$ brain area in BI2536-treated embryos increased significantly over time if compared to controls (p=2.2e-04 at stage 37/38, Wilcoxon rank-sum test). (d) Immunohistochemistry of nuclei (green) in whole-mount mitotic inhibitor-treated *Xenopus* embryo brains at successive developmental stages. Stages shown correspond approximately to the onset (stage 33/34), middle (stage 35/36), and end (stage 37/38) of tiv-AFM measurements. OT axons are outlined in purple. (e) Plots of the fold-change over time in stiffness gradient (orange) and OT turn angle (blue) for both control and mitotic inhibitor-treated embryos. Blocking mitosis significantly attenuated the rise in both stiffness gradient (p=0.01, linear regression analysis) and OT axon turning (p=0.02). Solid and dashed lines denote linear fits for control and inhibitor-treated embryos, respectively. (f) Time-lapse AFM montages showing fold-changes in brain stiffness in representative control (top; OT axons outlined in blue) and mitotic inhibitor-treated embryos (bottom; OT axons in magenta). The colour scale encodes the fold-change in $K$ at each location on the stiffness map, expressed relative to the values obtained at t = 0 min. (g) Boxplot of OT elongation at stage 37/38. Treatment with BI2536 significantly reduced OT elongation compared to controls (p=0.001, Wilcoxon rank-sum test). Boxplots show median, first, and third quartiles; whiskers show the spread of the data; 'o' indicates the mean. Error bars denote standard error of the mean. *p<0.05, **p<0.01, ***p<$10^{-3}$. AFM measurement resolution, 20 $\mu m$; all scale bars, 100 $\mu m$. N denotes number of animals except in (b) and (c) where it denotes number of sections.

DOI: https://doi.org/10.7554/eLife.39356.010

The following figure supplements are available for figure 4:

**Figure supplement 1.** The mitotic inhibitor BI2536 decreases nuclear density in *Xenopus* brains in vivo.
DOI: https://doi.org/10.7554/eLife.39356.011

**Figure supplement 2.** In vivo treatment with hydroxyurea/aphidicolin (HUA) reduces cell body density, OT elongation, and brain stiffness.
DOI: https://doi.org/10.7554/eLife.39356.012

**Figure supplement 3.** The impact of the mitotic inhibitor BI2536 on in vitro axon mechanosensing.
DOI: https://doi.org/10.7554/eLife.39356.013

---

*Figure 4—figure supplement 3d*). However, similar to control conditions, axons exposed to different concentrations of BI2536 grew significantly longer on stiff substrates compared to softer substrates. Hence, RGC axons responded to substrate stiffness despite the presence of the mitotic blocker (*Figure 4—figure supplement 3g*).

Our data show that, during early embryonic development, local tissue stiffness may change significantly within only tens of minutes (*Figure 2*), leading to heterogeneous stiffness distributions which, in the developing *Xenopus* brain, impact RGC axon growth. These stiffness heterogeneities were largely governed by differential cell proliferation (*Figures 3* and *4*), which is in agreement with previously published correlations between cell body densities and tissue stiffness (*Barriga et al., 2018*; *Koser et al., 2016*; *Koser et al., 2015*; *Weber et al., 2017*). Mitotic blockers almost completely abolished the rise of the stiffness gradient (*Figure 4*), which was accompanied by a decrease in turning angle of the OT, emphasizing the importance of the control of local cell proliferation for mechanosensitive cellular processes in vivo.

As changes in substrate stiffness have been shown to promote cell proliferation *in vitro* (*Georges and Janmey, 2005*), an increase in cell density might lead to a mechanical positive feedback loop, facilitating further cell proliferation. Perturbing cell division, on the other hand, might alter not only local tissue stiffness but also topological cues in the tissue, which may also provide important signals regulating axon growth. Having fewer cell bodies rostral to the OT might decrease the amount of steric hindrance and provide more space for axons to grow into, contributing to the reduction in OT turning angle (*Figure 4e*).

Our analysis of the correlation between local cell body density gradients and stiffness gradients suggests that other structures may contribute to the stiffness gradient as well (*Figure 3c*), although perhaps to a smaller degree. Potential candidates include radial glial cells (*MacDonald et al., 2015*) as well as components of the extracellular matrix (*Moeendarbary et al., 2017*).

Tiv-AFM allows simultaneous time-lapse measurements of tissue mechanics in vivo and optical monitoring of fluorescently labelled structures at the surface of otherwise optically opaque samples, at length and time scales that are relevant for developmental processes. It enabled us for the first time to trace the in vivo mechanical properties of the embryonic *Xenopus* brain as the embryo

developed, and to relate changes in tissue mechanics to a key event in axon pathfinding. As in all other AFM applications, in tiv-AFM forces are applied to sample surfaces, restricting it to biological processes that occur within tens of micrometers away from the surface, such as OT elongation in developing *Xenopus* embryos. Mapping of the whole *Xenopus* brain at cellular resolution takes about half an hour, which defines the maximum temporal resolution that can currently be achieved. Future developments of alternative approaches may enable measurements within tissues at even higher rates. However, tiv-AFM in its current form revealed that significant changes in tissue stiffness occur in vivo within tens of minutes, and that these changes have significant implications for a biological process, the outgrowth of RGC axons along the developing embryonic brain.

More broadly, tiv-AFM can also be easily adapted for in vivo applications in other small organisms, or alternatively in tissues ex vivo. It can be used to study cellular responses to a range of mechanical stimuli via the AFM in vivo, such as sustained compression (*Barriga et al., 2018*; *Koser et al., 2016*), or to track the temporal mechanical response of tissues or organs to different pharmacological treatments (such as the mitotic inhibitor used here). Additionally, the setup is very versatile and can be further expanded, for example, by combining it with calcium imaging to investigate how cellular activity is regulated by changes in tissue stiffness during development and pathology. Tiv-AFM will greatly expand the range of bio-AFM experiments possible, allowing for more scope both for a detailed characterisation of in vivo tissue mechanics during development and disease progression, and for testing how mechanics shapes cell behaviour and function.

## Materials and methods

All chemicals and reagents were obtained from Sigma-Aldrich unless otherwise noted.

### Key resources table

| Reagent type. | Designation. | Source (public). | Identifiers. | Additional information. |
|---|---|---|---|---|
| Transfected construct (*Xenopus laevis*) | ath5::GAP-eGFP | *Das et al., 2003*, PMID: 12597858; *Zolessi et al., 2006*, PMID: 17147778 | ZFIN ID: ZDB-TGCONSTRCT-070129–1 | Membrane-tagged GFP under control of ath5 (atoh7) promoter; pCS2 + vector |
| Biological sample (*Xenopus laevis*) | *Xenopus laevis* | National Xenopus Resource | Cat #: NXR_0.0031; RRID:SCR_013731 | Wild-type strain Xla. NXR-WT[NXR] |
| Antibody | Rabbit polyclonal anti-phospho-Histon H3 (sER10) | EMD Millipore | Cat #: 06–570; RRID:AB_310177 | IHC (1:1000) |
| Antibody | Goat anti-rabbit Alexa Fluor 594 | Abcam | Cat #: AB150084; RRID:AB_2734147 | IHC (1:500) |
| Chemical compound, drug | BI2536 | MedChem Express | Cat#: HY-50698 | 50 µM |
| Chemical compound, drug | Hydroxyurea | Sigma | Cat#: H8627-5G | 20 mM |
| Chemical compound, drug | Aphidicolin | Sigma | Cat#: A0781-5MG | 150 µM |
| Chemical compound, drug | 1M Sodium hydroxide | Sigma | | fabrication of polyacrylamide substrates |
| Chemical compound, drug | (3-aminopropyl) trimethoxysilane | Sigma-Aldrich | Cat#: 281778–100 ML | fabrication of polyacrylamide substrates |
| Chemical compound, drug | glutaraldehyde | Sigma-Aldrich | Cat#: G6257-1L | fabrication of polyacrylamide substrates |
| Chemical compound, drug | 40% w/v acrylamide solution | Sigma-Aldrich | Cat#: A4058-100ML | fabrication of polyacrylamide substrates |
| Chemical compound, drug | 2% bis-acrylamide | Fisher Scientific | Cat#: BP1404-250 | fabrication of polyacrylamide substrates |
| Chemical compound, drug | ammonium persulfate | Sigma | Cat#: 215589 | fabrication of polyacrylamide substrates |

*Continued on next page*

*Continued*

| Reagent type. | Designation. | Source (public). | Identifiers. | Additional information. |
|---|---|---|---|---|
| Chemical compound, drug | N,N,N',N'-tetramethylethylenediamine | ThermoFisher | Cat#: 15524–010 | fabrication of polyacrylamide substrates |
| Chemical compound, drug | hydrazine hydrate | Sigma-Aldrich | Cat#: 225819–500 mL | fabrication of polyacrylamide substrates |
| Chemical compound, drug | 5% acetic acid | ACROS Organics | Cat#: 10041250 | fabrication of polyacrylamide substrates |
| Chemical compound, drug | Poly-D-lysine | Sigma-Aldrich | Cat#: P6407-5MG | 10 µg/ml |
| Chemical compound, drug | laminin | Sigma-Aldrich | Cat#: L2020-1MG | 5 µg/ml |
| Software, algorithm | Fiji | Fiji is Just ImageJ (https://fiji.sc) | RRID:SCR_002285 | |
| Software, algorithm | Adobe Illustrator | Adobe Illustrator | RRID:SCR_010279 | |
| Software, algorithm | MATLAB | MATLAB | RRID:SCR_001622 | Codes used for Sholl analysis post-processing, OT elongation, motorized stage control, processing of AFM raw data, mapping of stiffness maps onto brains and OT and local tissue stiffness gradient calculations can be found at https://github.com/FranzeLab/AFM-data-analysis-and-processing (*Franze, 2018a*), https://github.com/FranzeLab/Image-processing-and-analysis (*Franze, 2018b*) and https://github.com/FranzeLab/Instrument-Control (*Franze, 2018c*; copies archived at https://github.com/elifesciences-publications/Image-processing-and-analysis, https://github.com/elifesciences-publications/Instrument-Control and https://github.com/elifesciences-publications/AFM-data-analysis-and-processing). |
| Other | 4,6-diamidino-2-phenylindole (DAPI) | Sigma-Aldrich | Cat#: D9542- 5 mg | 1 µg/ml |
| Other | Arrow-TL1 Tipless silicon cantilevers | NanoWorld Innovative technologies | Manufacturer's ID: ARROW-TL1-50 | Cantilevers for AFM-based stiffness measurements |
| Other | 37.28 µm spherical polysterene beads | microParticles GmbH | Cat#: PS-R-37.0 | Spherical probes attached to AFM cantilevers |
| Other | Rain-X | Shell Car Care International Ltd, UK | Model #: 800002250 | fabrication of polyacrylamide substrates |
| Other | CellHesion-200 AFM head | JPK Instruments | | Atomic force microscope |
| Other | PetriDish Heater | JPK Instruments | | Maintaining constant temperature for time-lapse AFM experiments |

## Animal model

All animal experiments were approved by the Ethical Review Committee of the University of Cambridge and complied with guidelines set by the UK Home Office. Single-cell-stage, wild-type *Xenopus laevis* embryos of both sexes were obtained via in vitro fertilisation. Embryos were reared in 0.1× Modified Barth's Saline (MBS) at 14–18°C to reach the desired developmental stage, as described by *Nieuwkoop and Faber, 1958*. All embryos used in this study were below stage 45.

### In vivo fluorescence labelling of optic tract (OT) axons.

To visualise the developing OT during time-lapse AFM experiments, an *ath5::GAP-eGFP* construct (pCS2$^+$ vector) (*Poggi et al., 2005*; *Das et al., 2003*) was injected (100 pg/5 nL) into one dorsal blastomere of embryos at the 4 cell stage. The construct consisted of a membrane-tagged GFP fusion under control of the retinal ganglion cell (RGC)-specific *atonal homolog 5* promoter (*Kanekar et al., 1997*). This selectively labelled RGCs in a single retina, the axons of which grew across the optic chiasm and into the unlabelled brain hemisphere.

### Exposed brain preparation

Stage 33/34 embryos were anaesthetised, the eye primordium was removed and the underlying brain hemisphere exposed as described (*Chien et al., 1993*; *Irie et al., 2002*). Briefly, embryos were transferred to 1.3 × MBS solution (composition: 1.3 × MBS with 0.04% (w/v) MS222 anaesthetic (3-aminobenzoic acid ethyl ester methanesulfonate) and 1 × penicillin/streptomycin/Fungizone (P/S/F; Lonza), pH 7.4). The higher osmolarity retards skin regrowth, allowing for experiments spanning several hours. Embryos were immobilised on low Petri dishes (TPP, Switzerland) coated with Sylgard 184 using bent 0.2 mm minutien pins, with one side of the body facing up. The eye, epidermis, and dura were removed with 0.1 or 0.15 mm minutien pins and fine forceps to expose one brain hemisphere from the dorsal to ventral midline and from the hindbrain to the telencephalon. Embryos were then immediately used for time-lapse in vivo AFM (tiv-AFM) measurements or, alternatively, transferred to a 4-well plate containing either 1.3× MBS solution+50 μM BI2536 (MedChem Express) (control, 1.3× MBS solution + 0.5% v/v DMSO) or 1.3× MBS solution + 20 mM hydroxyurea and 150 μM aphidicolin (control, 1.3× MBS solution + 1.5% v/v DMSO) and allowed to develop at ~25°C until the desired developmental stage. Embryo viability throughout all in vivo experiments was assessed by the presence of a visible heartbeat (which begins at st. 33/34 (*Gurdon et al., 1997*)).

### Cryosectioning Xenopus embryos

BI2536 inhibitor or mock-treated embryos were fixed at the requisite stages in 4% PFA overnight at 4°C, washed thrice in phosphate-buffered saline (PBS) for 10 min, and kept in 30% sucrose for 1 hr at 4°C. The embryos were embedded in optimum cutting temperature compound (OCT, VWR). 12μm-thick coronal sections were made and collected on Superfrost plus slides (ThermoScientific).

## In vitro assays

### Fabrication of polyacrylamide hydrogel substrates

Polyacrylamide hydrogels were prepared as previously described (*Koser et al., 2016*). Briefly, 19 mm 'bottom' coverslips were coated with 1N NaOH using a cotton bud and allowed to air dry. Coverslips were treated with (3-aminopropyl)trimethoxysilane (APTMS) for 3 min, washed thoroughly, treated with 0.5% glutaraldehyde solution for 30 min, and then washed and allowed to air-dry. 18 mm 'top' coverslips were treated with Rain-X (Shell Car Care International Ltd, UK) for 10 min and then dried.

Gel pre-mixes were prepared using 40% (w/v) acrylamide (AA) solution and 2% bis-acrylamide (Bis-AA) solution (Fisher Scientific, UK, or SIGMA) diluted in PBS. Concentration titration measurements used a premix composition previously determined to give a shear modulus *G* (a measure of stiffness) of ~300 Pa (5% AA, 0.07% Bis-AA in PBS). For gels used for stiffness sensing experiments, the precise stiffness was measured using AFM. Stiff gels were comprised of 7.5% AA/0.2% Bis-AA in PBS, resulting in a shear modulus of *G* ~5,500 Pa; soft gels were comprised of 5% AA/0.04% Bis-AA in PBS, resulting in *G* ~200 Pa.

Premix polymerization was initiated by adding 5 μL ammonium persulfate followed by 1.5 μL of N,N,N',N'-tetramethylethylenediamine (TEMED, ThermoFisher). 25 μL of premix was pipetted onto the bottom coverslip and the top coverslip placed on top. Once the gel had polymerized, the top coverslip was removed and gels were treated with hydrazine hydrate for 3.5–4 hr, and then with 5% acetic acid (ACROS Organics) for 1 hr. Gels were then washed, sterilized by 30 min UV treatment, and functionalized with 10 μg/mL Poly-D-lysine (MW 70,000–150,000) overnight followed by 5 μg/mL laminin for 2 hr immediately prior to plating cells.

## Xenopus tissue culture and in vitro inhibitor treatments

For eye primordia culture experiments, stage 33/34 or 35/36 embryos were placed in a Petri dish coated with Sylgard 184 (Dow Corning) and anaesthetized with 0.04% (w/v) MS222 solution (dissolved in $1 \times$ MBS + 1% v/v PSF, adjusted to pH 7.6–7.8, and filter-sterilized). Whole eye primordia were dissected out using insect pins, and placed onto hydrogels with the lens facing up. BI2536 or control solution (DMSO) was added 2 hr after explants were plated. Dishes were cultured at 20°C for 22–24 hr in *Xenopus* cell culture medium (60% L15 medium + 1× PSF, adjusted to pH 7.6–7.8, and filter sterilized). In all experiments, DMSO controls utilized the amount of DMSO equivalent to that in the most concentrated BI 2536 condition (0.1% v/v for concentration titration, 0.01% v/v for stiffness sensing experiments). Explants were imaged on a Leica DMi8 inverted microscope with a 10× NA = 0.4 phase contrast objective.

## Sholl analysis of RGC axon outgrowth

Eye primordia explant morphology was analyzed using the Sholl Analysis plugin in Fiji (*Ferreira et al., 2014*). An ellipse was fitted to the explant, and the innermost (starting) radius set to $R = \sqrt{A/\pi}$, with A being the ellipse area. Images were filtered with an FFT bandpass filter to correct uneven background illumination and manually thresholded. The outer radius was set to a point beyond the reach of the longest axon. Spacing between consecutive measurements was set to 5 µm. 'Median sholl radius' was calculated as the median outgrowth radius reached by axons of a particular explant.

## Fluorescence labelling of cellular structures

### Visualization of the optic tract and nuclei in wholemount brains

Where required, to visualise the OT and nuclei for cell body density measurements, embryos at the desired developmental stage were fixed in 4% paraformaldehyde for 1.5–2 hr at room temperature or overnight at 4°C. OTs were either labelled with *ath5::GAP-eGFP* or by injecting a solution of DiI crystals diluted in ethanol at the boundary between lens and the retina (as previously described *Wizenmann et al., 2009*). Fixed brains were then dissected out and stained with 4,6-diamidino-2-phenylindole (DAPI; 1 µg/ml). Stained specimens were mounted in either Fluoromount-G (eBioscience, UK; *ath5::GAP-eGFP*-labelled OTs) or $1 \times$ PBS (DiI-labelled OTs) and the lateral brain surface, including the OT, was imaged using an SP-8 confocal microscope (SP8, Leica Microsystems, UK; 20× air, NA = 0.75; z-step size = 1 µm).

### Phospho-histone H3 immunolabelling and imaging in tissue cross-sections

Sectioned tissues on slides were washed thrice in PBS, followed by three 10 min washes in PBS with 0.1% TritonX. The sections were blocked in 5% goat serum in PBS with 0.1% TritonX for 30–45 min and incubated with Rabbit polyclonal anti-phospho-Histone H3 (Ser10) (EMD Millipore, 06–570, dilution = 1:1000 in blocking solution) overnight at 4°C or for 2 hr at room temperature. This was followed by three 10 min washes in PBS and secondary antibody incubation with goat anti-rabbit Alexa Fluor 594 (Abcam, ab150084, dilution = 1:500 in blocking solution) for 45–60 min. The slides were washed twice for 10 min with PBS and nuclei were labeled using 4,6-diamidino-2-phenylindole (DAPI, 1 µg/ml). Sections were mounted with Fluoromount-G (eBioscience) and imaged with a confocal microscope (SP8, Leica Microsystems, UK; 20×/0.75 air and 63×/1.4 oil). Z-stacks were taken across 8 µm of tissue (z-step size = 1 µm). Only slices with eye tissue present were selected, to reliably ensure that the brain sections imaged and analysed were indeed exposed to the treatment solutions. This also allowed us to easily ascertain the intact- versus exposed- brain side in each section.

## In vivo AFM

### Probe and instrument preparation

Tipless silicon cantilevers (Arrow-TL1, NanoWorld) were calibrated using the thermal noise method (*Hutter and Bechhoefer, 1993*) to determine the spring constant $k$, and those with $k$ between 0.02–0.04 N/m were selected. Monodisperse spherical polystyrene beads (diameter 37.28 ± 0.34 µm; microParticles GmbH) were glued to the cantilever ends as probes. Cantilevers were mounted on a CellHesion-200 AFM head (JPK Instruments), which was set up on an x/y motorised stage (JPK

Instruments) controlled by custom-written Python scripts (*Koser et al., 2016*). Indentation measurements (maximum indentation force: 10 nN, approach speed: 5 µm/s, data rate: 1,000 Hz) were performed automatically in a user-defined rectangular grid to create a 2-D 'stiffness map' of the area. After each measurement, the cantilever was retracted by 100 µm, and the stage moved by a set distance (20–25 µm) to the next position (*Koser et al., 2016*).

## Combined optical imaging and AFM measurements

To allow simultaneous time-lapse imaging of the growing OT and in vivo stiffness measurements of live *Xenopus* brains, which are optically opaque, an upright epifluorescence set-up was custom-fit above the AFM head, which was based on a Zeiss AxioZoom V16 fluorescence stereomicroscope (without eyepieces). The microscope was mounted on a custom-designed horizontal sliding stand (parts obtained from ThorLabs) via a custom-made adaptor (K-Tec Microscope Services) bolted to the side of the microscope (*Figure 1—figure supplement 1*). A long working distance objective with automatic zoom and high NA (PlanApo Z 0.5×/0.125, working distance 114 mm, Zeiss) allowed in vivo fluorescence imaging of RGC axons through the transparent AFM head, which was re-fitted with optical elements optimised for fluorescence imaging. The sample was illuminated through the AFM head using a metal halide lamp, and bright-field as well as epifluorescence images of the brain collected with an sCMOS camera (Andor Zyla 4.2; quantum efficiency 82%) (*Figure 1a,b*).

## Tiv-AFM

Stage 33/34 embryos with *ath5::GAP-eGFP*-labelled OTs were prepared with one brain hemisphere exposed as described above, and those in which the OT was clearly visible (but had not yet formed the mid-OT bend) were selected. Embryos were mounted on the AFM motorised stage and a measurement region of approximately 150 × 250 µm defined to include both the growing OT and the region of the mid-diencephalic turn. Images of the upper right and lower left corners of the selected region (with cantilever approached) were collected to identify the precise area mapped by the AFM. A stiffness map of the area was collected and the map iterated over the same area every ~35 min. At the end of every line in the measurement grid, the stage was moved back to a pre-defined location and a fluorescence image of the optic tract automatically collected. Temperature was maintained at 25°C for the duration of the measurement by a PetriDish Heater (JPK Instruments).

## Data analysis

### Processing of raw AFM data

Force-distance curves obtained from stiffness measurements were analysed with a custom-written MATLAB script (*Koser et al., 2016*; *Christ et al., 2010*) to obtain the reduced apparent elastic modulus *K*. Raw AFM data were fitted to the Hertz model,

$$\mathbf{F} = \frac{4}{3}K\delta^{\frac{3}{2}}\sqrt{\mathbf{R}}$$

where *F* is the applied force, *K* the reduced apparent elastic modulus $E/(1-\nu^2)$ (with Poisson ratio ν), *R* the radius of the indenter, and δ is sample indentation (*Crick and Yin, 2007*; *Hertz, 1882*). Force-distance curves were analysed at the maximum applied force *F* = 10 nN. Points where the AFM data were not analysable were excluded; criteria for excluding individual force-distance curves were (1) inability to apply linear fits through the baseline region, for example due to noise, and (2) inability to apply good-quality Hertzian fits to the indentation region, that is, the fit did not align with the raw data.

To minimise noise for region-of-interest analysis of small areas, previously sorted, gridded AFM data were smoothed in x-, y-, and time dimensions using an algorithm based on the discrete cosine transform by Garcia et al (*Garcia, 2010*; *Garcia, 2011*). Briefly, an iteratively weighted version of the penalized least squares approach was used to smooth the data and interpolate missing values where the force-distance curves were not analysable. Stiffness data from each experiment were arranged into a 3D array (x*y*frame number) and used as an argument for the *smoothn.m* MATLAB implementation of the algorithm (*Garcia, 2010*). This algorithm has the advantage of using the entire data set to interpolate missing values, as opposed to only the nearest neighbouring data (as, for example, in other smoothing methods such as a simple spline interpolation).

For visual presentation of interpolated stiffness data, additional arrays of robustly smoothed $K$ values were generated using the separate 'robust' option available in *smoothn.m*. This automatically reduces the weight assigned to high-leverage points and outliers, which optimises visualisation of overall changes in stiffness. To generate time-lapse montages of both raw and robustly smoothed AFM data, values of $K$ were converted to 8-bit scale, colour-coded maps using the desired MATLAB colourmap pre-set. The resulting stiffness maps were then mapped onto images of the brain and OT using custom-written MATLAB scripts (*Koser et al., 2016*).

## Quantifying in vivo stiffness gradients

Another MATLAB script (*Koser et al., 2016*) was used to calculate local tissue stiffness gradients in the area just ahead of the advancing OT. Mean values of $K$ were calculated for a $50 \times 50$ µm$^2$ region of interest (ROI) on the rostral ($K_R$) and caudal sides ($K_C$) of the OT, and the gradient calculated by:

$$Stiffness\ gradient = \frac{K_R - K_C}{50\ \mu m}$$

For each frame in a given time-lapse series, ROIs were defined three times and the mean of the three measurements for each ROI was used for further analysis. Where direct comparison between the dynamics of stiffness gradients and OT turning was required across different animals, values for both parameters were rescaled. The minimum value obtained for each embryo measured was set to 0 and the maximum set to 1, using the following formula:

$$Rescaled\ data = \frac{data - minimum\ value}{maximum\ value - minimum\ value}$$

When linear fits to AFM data and OT turning angles were required, the built-in MATLAB first degree polynomial fit was used in all cases.

## Analysis of OT turning

For each embryo, in vivo time-lapse movies of the growing OT were created by collating fluorescence images taken at the beginning of every successive AFM stiffness map. The magnitude of the OT turn was calculated from each frame of these movies using the FIJI Angle tool. Three points were defined manually (chiasm, mid-diencephalic turn, and end of OT) such that the two lines drawn through these points ran through the centre of the OT. Each measurement was repeated three times, averaged, and subtracted from 180° to give the angle through which the OT had turned; a positive value denotes turning in the caudal direction.

## Analysis of OT elongation

Maximum projections were made across 10 µm confocal image stacks of wholemount brains with DiI-labelled OTs. The OTs were manually outlined in Adobe Illustrator. Elongation of the OT was calculated by the major-to-minor axis ratio using a previously described automated algorithm in Matlab (*Koser et al., 2016*). In short, axes were determined by fitting ellipses, with the same normalized second central moment as the OT area, around the OTs.

## Cell body density measurements

Confocal image stacks of fixed brains (with fluorescently-labelled OT and nuclei) were imported into FIJI. For each brain, the image where the leading axons were in focus was selected. A maximum intensity projection was generated of this image and one image above and below it in the stack. Two 50 µm × 50 µm regions of interest (matching the size of regions of interest used in stiffness gradient analysis) were selected, rostral ($D_R$) and caudal ($D_C$) to the region corresponding to the OT caudal bend. Noise was removed with a Gaussian blur filter (sigma = 2.0). The resulting image was thresholded, with the threshold manually adjusted to capture all nuclei as accurately as possible. The image was binarised and the FIJI built-in function 'Analyse Particle' (size: 1–∞; circularity: 0.2–1.00) was used to determine the area in each region covered by nuclei. Relative cell body density was calculated by dividing the area occupied by nuclei by the total region area. Where required, gradients in relative cell body density were obtained using:

$$Cell\ body\ density\ gradient = \frac{D_R - D_C}{50\ \mu m}$$

## Measurement of cross-sectional brain area

Confocal images taken with the 20 × air objective were imported into FIJI. Maximum projections were made of the stacks (z-stack height = 8 μm). The brain was identified in each section and both intact and exposed hemispheres were outlined. The area occupied by nuclei in each brain hemisphere was measured using the 'Analyse > Measure' tool. Normalised brain areas were obtained by calculating the ratio of exposed brain area to intact brain area.

## Quantification of phospho-histone H3 immunolabeling

Confocal images taken with the 20 × air objective were imported into FIJI. Maximum projections were made of the stacks (z-stack height = 8 μm) and pH3+ cells were counted and normalized to the total measured brain area in each section. The data is presented as the number of pH3+ cells per 10,000 μm² of brain tissue.

$$\frac{number\ of\ pH3 + cells}{brain\ area\ (\mu m^2)} * 10,000$$

### Statistics and visualisation

Data were collected from at least three independent experiments (N ≥ 3). The order of data collection was randomized with no blinding and no data were excluded from the analysis. Non-parametric tests as well as linear regression analysis were used for statistical analyses of the data as described in the figure captions. $R^2$ values provide an estimate of the quality of the fits used in the plots. Pearson's correlation coefficient, on the other hand, provides a measure of the magnitude of correlation between the cell body density gradient and the stiffness gradient.

## Acknowledgements

The authors would like to thank Alex Winkel (JPK Instruments), Katrin Mooslehner, Asha Dwivedy, Liz Williams, and Julia Becker for technical help, Alex Robinson and Max Jakobs for scientific discussions and logistical support, as well as the Wellcome Trust (grant 099743/Z/12/Z to AJT), the Cambridge Philosophical Society and the Cambridge Trusts (studentships to AJT), the Malaysian Commonwealth Studies Centre (funding to EKP), the UK Engineering and Physical Sciences Research Council Cambridge NanoDTC (grant EP/L015978/1 to IBD), the Herchel Smith fund (studentship to SKF), the European Research Council (Advanced Grant 322817 to CEH and Consolidator Award 772426 to KF), the UK Biotechnology and Biological Sciences Research Council (BB/M020630/1 and BB/N006402/1 to KF), the UK Medical Research Council (Career Development Award G1100312/1 to KF), and the Eunice Kennedy Shriver National Institute Of Child Health and Human Development of the National Institutes of Health under Award Number R21HD080585 (to KF) for funding support. The content is solely the responsibility of the authors and does not necessarily represent the official views of the National Institutes of Health.

## Additional information

### Funding

| Funder | Grant reference number | Author |
| --- | --- | --- |
| Wellcome Trust | 099743/Z/12/Z | Amelia J Thompson |
| Engineering and Physical Sciences Research Council | EP/L015978/1 | Ivan B Dimov |
| Herchel Smith Foundation | Research Studentship | Sarah K Foster |
| European Research Council | 322817 | Christine E Holt |
| Biotechnology and Biological Sciences Research Council | BB/M020630/1 | Kristian Franze |

| Medical Research Council | G1100312/1 | Kristian Franze |
|---|---|---|
| Eunice Kennedy Shriver National Institute of Child Health and Human Development | R21HD080585 | Kristian Franze |
| European Research Council | 772426 | Kristian Franze |
| Biotechnology and Biological Sciences Research Council | BB/N006402/1 | Kristian Franze |

The funders had no role in study design, data collection and interpretation, or the decision to submit the work for publication.

### Author contributions

Amelia J Thompson, Eva K Pillai, Conceptualization, Data curation, Formal analysis, Validation, Investigation, Visualization, Methodology, Writing—original draft, Writing—review and editing; Ivan B Dimov, Software, Formal analysis, Methodology, Writing—review and editing; Sarah K Foster, Data curation, Formal analysis, Validation, Investigation, Writing—review and editing; Christine E Holt, Resources, Supervision, Writing—review and editing; Kristian Franze, Conceptualization, Supervision, Funding acquisition, Investigation, Visualization, Methodology, Writing—original draft, Project administration, Writing—review and editing

### Author ORCIDs

Amelia J Thompson (iD) https://orcid.org/0000-0002-3912-3652
Kristian Franze (iD) http://orcid.org/0000-0002-8425-7297

### Decision letter and Author response

Decision letter https://doi.org/10.7554/eLife.39356.018
Author response https://doi.org/10.7554/eLife.39356.019

## Additional files

### Supplementary files

• Transparent reporting form
DOI: https://doi.org/10.7554/eLife.39356.014

### Data availability

All data generated or analysed during this study are included in the manuscript and supporting files. Codes used for Sholl analysis post-processing, OT elongation, motorized stage control, processing of AFM raw data, mapping of stiffness maps onto brains and OT and local tissue stiffness gradient calculations can be found at https://github.com/FranzeLab/AFM-data-analysis-and-processing, https://github.com/FranzeLab/Image-processing-and-analysis and https://github.com/FranzeLab/Instrument-Control (copies archived at https://github.com/elifesciences-publications/Image-processing-and-analysis, https://github.com/elifesciences-publications/Instrument-Control and https://github.com/elifesciences-publications/AFM-data-analysis-and-processing).

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
