## [Decision Letter]

Thank you for submitting your article "Rapid changes in tissue mechanics regulate cell behaviour in the developing embryonic brain" for consideration by *eLife*. Your article has been reviewed by three peer reviewers, and the evaluation has been overseen by a Reviewing Editor and Marianne Bronner as the Senior Editor. The following individuals involved in review of your submission have agreed to reveal their identity: Paola Bovolenta (Reviewer #1); Valerie Castellani (Reviewer #2).

The reviewers have discussed the reviews with one another and the Reviewing Editor has drafted this decision to help you prepare a revised submission.

The three reviewers appreciated your efforts to probe how dynamic changes in tissue mechanics have a close relationship with developmental processes, here, with a focus on axonal growth. Your findings that the pathway that retinal ganglion cell axons follow in the *Xenopus* brain is influenced by relatively rapid changes in tissue stiffness on which the axons grow, and that increasing stiffness correlates with enhanced cell density/proliferation, are indeed novel. Your approach of simultaneously observing and quantifying tissue stiffness and cell movements using atomic force microscopy with fluorescence imaging is exciting and will appeal to the growing number of groups interested in tissue morphogenesis.

Essential revisions:

These experiments are considered necessary to strengthen the connection you make between proliferation, density, and tissue stiffness:

1) Coronal sections of control and BI2536 treated embryos showing nuclei (DAPI stained) and glial endfeet (GFAP or Nestin labelling).

2) Labelling sections with pH3 or other mitotic cell markers.

3) Effects of the BI2536 inhibitor: Additional movies are needed (Figure 2F, N= only 2 for the BI2536 condition).

4) Check whether RGC axons grow and sense stiffness normally in vitro in presence of BI2536 (for example, using the axon outgrowth assay described in Koser et al., 2016).

5) Calculate optic tract (OT) elongation in control and BI2536-treated conditions. Related to BI2536 treatment and turning in the optic tract: does BI2536 generally affect axon growth along the optic tract in vivo?

6) Demonstrate that repeated AFM measurements do not affect OT turning (by comparing with movies without AFM measures).

7) Confirm with a second drug that cell density controls brain stiffness and RGC axon turning.

The following points do not rely on additional experiments but should be addressed:

1) State in the text the optical resolution and the accuracy of the AFM measure relative to the surface distance (in other words, is AFM producing a measure of the stiffness of the tissues 10μm or 100μm below the surface scanned by the probe?).

2) Provide illustrations of the angle measurements (in the video in Figure 1, it seems that the turn is initiated at 37 minutes).

3) Show the ROIs used to quantify the stiffness gradient and the boxplot of the mean K values in rostral and caudal regions at different time points. In addition, using different ROIs along the OT would strengthen the specificity of the correlation between stiffness gradient and turning.

4) Show several examples of the stiffness changes to better appreciate the spatial stereotypy relative to OT development.

5) Figure 1D, related to text "a stiffness gradient arose, mostly due to rising stiffness of tissue rostral to the OT". Some of the heatmap data (e.g., Figure 1—figure supplement 2C) suggests that there is decreasing stiffness on the caudal side, and relative cell density data (Figure 2D) suggests that there is decreasing cell density on the caudal side as well. Can the authors comment on the rostral-caudal stiffness v cell density? Is there some underlying biology to account for this?

6) Explain why only 3 time points are presented in the graph Figure 2A and G even though measurements are made every 35 minutes.

7) Figure 2B: According to the graph, it appears that the stiffness gradient is projected to form at 8-10 minutes. This suggests that it may begin to form prior to the beginning of the experiments presented (if the gradient forms linearly at those earlier times). Have the authors tried starting the experiments earlier? The manner in which the stiffness gradient forms (linearly or non-linearly) may have implications for the mechanism(s) initiating and influencing stiffness.

8) Specify the estimated delay between gradient stiffness establishment and OT caudal turn (15-18 minutes). Paired analysis of the onset times of stiffness gradient and turn for each video (ladder graphic) would strengthen the arguments. It would be interesting to have an estimate of the stiffness differences that would lead to axon turning. This would be helpful together with point 4 to make predictions and to enrich the model proposed in Koser et al., 2016.

9) Show a 3D reconstruction of the Z-stack used for Figure 2C and specify the total stack height.

10) Explain how "cell body density gradients" were calculated (Figure 2E) (only "relative cell body density" is described in the Materials and methods). The authors should take into account that quantification made on a Z projection will lead to loss of information that will decrease the sensitivity of the analysis. Figure 2E suggests that a more balanced conclusion should be drawn from the experiment, as part of gradient stiffness is independent of cell density gradient (gradient stiffness would thus be 0.4 for no cell density gradient).

11) Provide relative cell body density for the BI2536 condition, as done in Figure 2D for the control (indeed, the authors stated that "cell density gradient was significantly decreased compare to controls, particularly at later stages (Figure 2C)").

12) The Materials and methods section indicates that relative density involves ratio of 2D area in the image occupied by nuclei over total area (this does not appear to be a cell count). This measurement could be affected by differences in nuclear size. Have the authors measured nuclear size over time? Does this change? Is nuclear size different in BI2536-treated embryos? Also, it is difficult to see nuclei on the caudal side of the optic tract in Figure 2C (control).

13) Again, related BI2536 and cell density measurements, over time for BI2536-treated embryos: It would be useful to know if the BI2536 caused apoptosis (leading to an overall reduction in density)? The nuclei look somewhat condensed on what appears to be the rostral side of the tract in the BI2536 sample by st. 37 (Figure 2C).

14) Show a different control movie in Figure 2F (OT trace looks the same as in Figure 1).

15) Justify the changes of presentation between Figure 2A and G.

16) Explain what BI2536 targets (inhibitor of polo-like kinase).

17) Explain differences between coefficient R^2^= 0.95 in Figure 2C and Pearsons coefficient in the text (p=0.97). Add scale bars in Figure 2C and F.

For consideration in the Discussion:

1) You suggest that the "gradient in cell densities is responsible for the stiffness gradient". However, the opposite could be true, as changes in stiffness have been shown to promote proliferation (should be referenced). In addition, as you previously showed that ECM manipulation changes the tissue stiffness and axon trajectory (Koser et al., 2016), it would be welcome to discuss whether you have assessed whether changes in cell density correlate with changes in ECM deposit and if BI2536 impairs it.

2) In the last paragraph, you discuss applications of tiv-AFM to other possible contexts. Limitations of the system should be also mentioned.

---

## [Author Response]

Essential revisions:These experiments are considered necessary to strengthen the connection you make between proliferation, density, and tissue stiffness:1) Coronal sections of control and BI2536 treated embryos showing nuclei (DAPI stained) and glial endfeet (GFAP or Nestin labelling).

We have done the suggested experiments as far as possible. DAPI staining of coronal sections confirmed a decrease in overall cell numbers after treatment with BI2536 (Figure 4A, B). However, GFAP is not expressed in *Xenopus* brains at the early developmental stages we were interested in (Messenger and Warner, Development, 1989; Martinez-De Luna et al., Dev Biol, 2017; and our own data). Furthermore, we could not find a Nestin antibody with sequence similarities above 85% with *Xenopus* (we tested two commercially available antibodies, Abcam ab134017 Chicken polyclonal to Nestin and ab81462 Rat monoclonal [7A3] to Nestin on coronal sections anyway, but with limited success). We also stained for vimentin, which is another glial cell marker. While we obtained a good signal from that approach, we could not distinguish individual glial cell endfeet. As we cannot make any strong conclusions about the contribution of glial cells to the stiffness gradients and axon turning based on these data (see Author response image 1), we now discuss their potential involvement in the main text.

**Author response image 1. respfig1:** Coronal brain sections of stage 37/38 *Xenopus* embryos stained for DAPI (red) and Nestin (above) or Vimentin (below) (white). Control animals (left) and BI2536-treated animals (right); the exposed side is on the right hand side of each image. All scale bars are 20 µm.

2) Labelling sections with pH3 or other mitotic cell markers.

We have now labelled coronal sections with an antibody to detect histone H3 phosphorylated at serine 10, a marker of mitotic cells. Ser10 phosphorylation begins in G2 and decreases upon exit from mitosis. We found that the BI2536 treatment successfully triggers mitotic arrest as the number of pH3-positive cells is significantly higher in treated compared to control brains. These data have been added to Figure 4A, C.

3) Effects of the BI2536 inhibitor: Additional movies are needed (Figure 2F, N= only 2 for the BI2536 condition).

We have repeated these experiments with 2 more biological replicates and updated the numbers in the paper (N = 4). The new sets of experiments fully agree with our previous data, confirming reproducibility of the experiments and that the stiffness gradients is almost completely abolished after BI2536 treatment (Figure 4E).

4) Check whether RGC axons grow and sense stiffness normally in vitro in presence of BI2536 (for example, using the axon outgrowth assay described in Koser et al., 2016).

We have done the suggested in vitroexperiments. Axons exposed to 50nM and 500nM BI2536 still sense and respond to the stiffness of their environment, suggesting that the main effect of the drug is through a modification of tissue mechanics. The new data have been added to Figure 4—figure supplement 3.

5) Calculate optic tract (OT) elongation in control and BI2536-treated conditions. Related to BI2536 treatment and turning in the optic tract: does BI2536 generally affect axon growth along the optic tract in vivo?

We have calculated OT elongation as suggested and find differences between controls and BI2536-ând HUA-treated animals. In the treated animals, elongation is significantly reduced, which we now show in Figure 4G and Figure 4—figure supplement 2D. As brain tissue stiffness is decreased in treated animals with decreased nuclear densities (Figure 4E, F, Figure 4—figure supplement 2E, F), which is in agreement with an important role of cell body density in regulating tissue stiffness, a reduction in OT elongation is in line with our previous publication (Koser et al., 2016).

6) Demonstrate that repeated AFM measurements do not affect OT turning (by comparing with movies without AFM measures).

To address this point, we conducted time-lapse AFM measurements (i.e. repeated AFM mapping over a time course of at least 3 hours) on a group of embryos while we exposed another group to the same conditions but did not do any force measurements. At the end of the experiments (i.e., at stage 37/38), embryos were fixed, optic tracts labelled with DiI, and their elongation and turning angles measured. We did not find any significant differences between the groups, suggesting that repeated AFM measurements do not affect OT turning. We have added the new data to Figure 2—figure supplement 2 and the fifth paragraph of the main text.

7) Confirm with a second drug that cell density controls brain stiffness and RGC axon turning.

We have now used a second mitotic inhibitor treatment, hydroxyurea/aphidicolin (HUA) (which has previously been used in *Xenopus* embyros in vivo: Harris and Hartenstein, 1991), which resulted in a very similar optic tract phenotype as seen in BI2536-treated brains, with significantly reduced OT elongation and brain stiffness. These experiments are now shown in Figure 4—figure supplement 2.

The following points do not rely on additional experiments but should be addressed:1) State in the text the optical resolution and the accuracy of the AFM measure relative to the surface distance (in other words, is AFM producing a measure of the stiffness of the tissues 10μm or 100μm below the surface scanned by the probe?).

AFM is applied to the surface of a sample. The measured stiffness is a composite property of structures within the top layers of the sample, with a larger contribution of structures that are closer to the surface. The total depth that contributes to the measured value depends on the mechanical properties of the sample, the force applied, and the contact area between probe and sample. In our experiments, we indent the samples by ~3-4 µm. We estimate structures to contribute to the measured stiffness mainly within the top ~20-30 µm (see simulation in Author response image 2), which includes the region through which RGC axons grow (Holt, J Neurosci, 1989). We now mention this in the fourth paragraph of the main text.

**Author response image 2. respfig2:** Simulation of the propagation of tissue deformation during an AFM measurement of a tissue with an elastic modulus of K = 300 Pa with an applied maximum force F = 10 nN and a 37 µm bead as probe. The sample is indented by 3.4 µm, tissue deformation propagates into deeper layers. Most contributions to the measured elastic modulus come from structures within the top ~20-30 µm, within which RGC axons grow.

2) Provide illustrations of the angle measurements (in the video in Figure 1, it seems that the turn is initiated at 37 minutes).

Done as suggested (Figure 1C).

3) Show the ROIs used to quantify the stiffness gradient and the boxplot of the mean K values in rostral and caudal regions at different time points. In addition, using different ROIs along the OT would strengthen the specificity of the correlation between stiffness gradient and turning.

We now show the ROIs used to quantify the stiffness gradient and boxplots of the median *K* values in rostral and caudal regions at different time points in Figure 2—figure supplement 1. However, we have chosen the ROIs to be directly in front of the advancing OT axons because this is the region that growth cones will be sensing. Therefore, we would rather not include different ROIs along the OT to avoid confusion, as it is not clear whether these would contribute to axon guidance in any way.

4) Show several examples of the stiffness changes to better appreciate the spatial stereotypy relative to OT development.

We now show two more examples of time-lapse stiffness maps in Figure 2—figure supplement 3.

5) Figure 1D, related to text "a stiffness gradient arose, mostly due to rising stiffness of tissue rostral to the OT". Some of the heatmap data (e.g., Figure 1—figure supplement 2C) suggests that there is decreasing stiffness on the caudal side, and relative cell density data (Figure 2D) suggests that there is decreasing cell density on the caudal side as well. Can the authors comment on the rostral-caudal stiffness v cell density? Is there some underlying biology to account for this?

We generally see a strong correlation between local cell body densities and tissue stiffness. Accordingly, tissue stiffness increases rostral to the OT, where cell density increases during development. Caudal to the OT, cell density decreases slightly in some but not all embryos, and accordingly we don’t see statistically significant differences in tissue stiffness caudal to the OT.

6) Explain why only 3 time points are presented in the graph Figure 2A and G even though measurements are made every 35 minutes.

Our aim was to match the mechanics measurements to the three established developmental stages of *Xenopus* embryos encompassing the duration of our experiments (stages 33/34, 35/36, and 37/38), in order to enable meaningful comparisons with local cell body densities, which were assessed at these stages. Therefore, values from stiffness maps had to be pooled. We now justify this procedure in the caption of Figure 2.

7) Figure 2B: According to the graph, it appears that the stiffness gradient is projected to form at 8-10 minutes. This suggests that it may begin to form prior to the beginning of the experiments presented (if the gradient forms linearly at those earlier times). Have the authors tried starting the experiments earlier? The manner in which the stiffness gradient forms (linearly or non-linearly) may have implications for the mechanism(s) initiating and influencing stiffness.

Our data indeed suggest that stiffness gradients may start forming a few minutes before the start of our measurements. However, our data strongly indicate that the stiffness gradient forms rather linearly during the following ~3 hours between stage 33/34, when the pioneering axons first leave the optic chiasm and enter the diencephalon, and stage 37/38, when axons have turned in the mid-diencephalon (Figure 2A). Measurements at earlier developmental stages would be technically highly challenging.

8) Specify the estimated delay between gradient stiffness establishment and OT caudal turn (15-18 minutes). Paired analysis of the onset times of stiffness gradient and turn for each video (ladder graphic) would strengthen the arguments. It would be interesting to have an estimate of the stiffness differences that would lead to axon turning. This would be helpful together with point 4 to make predictions and to enrich the model proposed in Koser et al., 2016.

These are indeed excellent suggestions. We now specify the delay of 18 minutes in the seventh paragraph of the main text and in Figure 2E. Furthermore, we include a ladder graphic as suggested, in addition to our paired analysis of the onset times, in Figure 2D. We now also include an estimate of the minimum stiffness gradient that would lead to axon turning, which is (0.9 ± 0.4) Pa/µm.

9) Show a 3D reconstruction of the Z-stack used for Figure 2C and specify the total stack height.

As mentioned in the Materials and methods, we only recorded three optical planes in each stack to make sure we capture all nuclei in that region. Hence, a 3D reconstruction would not add to the information that is already provided. Moreover, the coronal sections that we now added (see experiment 1 above, Figure 4) contain information about the depth-distribution of nuclei, which we hope addresses this point.

10) Explain how "cell body density gradients" were calculated (Figure 2E) (only "relative cell body density" is described in the Materials and methods). The authors should take into account that quantification made on a Z projection will lead to loss of information that will decrease the sensitivity of the analysis. Figure 2E suggests that a more balanced conclusion should be drawn from the experiment, as part of gradient stiffness is independent of cell density gradient (gradient stiffness would thus be 0.4 for no cell density gradient).

We apologise for this oversight and added an explanation of cell body density gradient calculations to the Materials and methods. As mentioned above, our Z projections consisted of three optical slices only that were spaced by 2 µm, so that we would expect the loss of information to be rather small.

The extrapolation of the fit in Figure 2E is an excellent point. When we force the fit through the coordinate origin (which would reflect a scenario in which the cell body density gradient is the only major contributor to the stiffness gradient), the R^2^ value is still 0.74. However, we agree with the reviewer that other tissue components might very well also contribute to setting up the stiffness gradient to some degree, and we now tone our conclusions down and mention potential other sources including radial glial cells and the extracellular matrix. We would like to point out, however, that even if other cellular and/or extracellular components are involved in setting up an initial stiffness gradient (which would not be strong enough to cause axon turning), our experiments with two different mitotic blockers strongly suggest that the cell body density gradient is responsible for the development of this stiffness gradient to the point where it impacts axon turning, as perturbations of mitosis abolish the stiffness gradient almost completely.

11) Provide relative cell body density for the BI2536 condition, as done in Figure 2D for the control (indeed, the authors stated that "cell density gradient was significantly decreased compare to controls, particularly at later stages (Figure 2C)").

We now provide a relative cell body density count for the BI2536 condition (Figure 4—figure supplement 1) and rephrased our conclusion: ‘The inspection of the stage-dependent distribution of cells in the developing *Xenopus* brain suggested that in BI2536-treated brains, the nuclear density decreased particularly rostral to the OT (Figure 4D, Figure 4—figure supplement 1).’

12) The Materials and methods section indicates that relative density involves ratio of 2D area in the image occupied by nuclei over total area (this does not appear to be a cell count). This measurement could be affected by differences in nuclear size. Have the authors measured nuclear size over time? Does this change? Is nuclear size different in BI2536-treated embryos? Also, it is difficult to see nuclei on the caudal side of the optic tract in Figure 2C (control).

The reviewer is correct, we intentionally quantified the area occupied by nuclei rather than their number. Nuclei are usually much stiffer than, for example, cytoplasm. From a mechanical point of view (under the assumption that nuclear stiffness does not significantly vary with volume but rather with lamin A/C content, see for example Swift et al., Science, 2013), it does not matter whether the total nuclear area increases because of a change in nuclear size or number. The number of nuclei caudal of the OT is indeed very low, which likely explains (at least to some degree) why the tissue in that region is softer than cranial of the OT, where a significantly larger volume is covered by nuclei.

13) Again, related BI2536 and cell density measurements, over time for BI2536-treated embryos: It would be useful to know if the BI2536 caused apoptosis (leading to an overall reduction in density)? The nuclei look somewhat condensed on what appears to be the rostral side of the tract in the BI2536 sample by st. 37 (Figure 2C).

We cannot rule out that BI2536 caused some degree of apoptosis at the application site. However, again, from a mechanical point of view it does not matter much why nuclear density changes. The scope of this paper is to show that local tissue stiffness may change rather rapidly during development, and that these changes are largely governed by changes in nuclear densities. We would like to point out that retinal ganglion cells, whose axons grow along the optic tract, are located within intact eye primordia and are thus not exposed to the drug in exposed brain preparations.

14) Show a different control movie in Figure 2F (OT trace looks the same as in Figure 1).

We intentionally showed the same data in Figures 1 and 2, as we wanted to make a comparison easier (as Figure 1 shows absolute stiffness and Figure 2 relative changes in stiffness). However, we see the value of showing different time lapse data and added two more series to the supplement (Figure 2—figure supplement 3).

15) Justify the changes of presentation between Figure 2A and G.

In Figure 2A (now Figure 2C), we show the stiffness gradients and turning angles normalised by the maximum values measured, which allowed us to extrapolate linear fits and determine the relative onset of both gradients and angles (leading straight to current Figures 2D, E). This normalisation is very useful as absolute values will vary slightly not only between individual animals but also between different AFM cantilevers. In Figure 2G (now Figure 3C), however, we wanted to compare treated embryos with controls. If we would follow the same normalisation process, the gradients of both groups would terminate at 1 and no differences would be visible. This problem was circumvented by normalising with respect to the first measurement in a time series (i.e., to the lowest value). This way, Figure 2G shows clearly that in controls but not in treated animals stiffness gradients and the turning angles are increasing over time.

16) Explain what BI2536 targets (inhibitor of polo-like kinase).

Done as suggested (main text, tenth paragraph).

17) Explain differences between coefficient R^2^= 0.95 in Figure 2C and Pearsons coefficient in the text (p=0.97). Add scale bars in Figure 2C and F.

The R^2^ value provides an estimate of the quality of fits to data distributions. Pearson’s correlation coefficient, on the other hand, provides a measure of the magnitude of correlation between the cell body density gradient and the stiffness gradient. We have added this information to the Methods. We have now increased the visibility of the scale bars in the figures.

For consideration in the Discussion:1) You suggest that the "gradient in cell densities is responsible for the stiffness gradient". However, the opposite could be true, as changes in stiffness have been shown to promote proliferation (should be referenced). In addition, as you previously showed that ECM manipulation changes the tissue stiffness and axon trajectory (Koser et al., 2016), it would be welcome to discuss whether you have assessed whether changes in cell density correlate with changes in ECM deposit and if BI2536 impairs it.

We would like to thank the reviewer for this comment. If cell density-independent changes in tissue stiffness would drive the growth of the stiffness gradient and independently lead to differential cell proliferation, interfering with cell proliferation should not diminish the stiffness gradient. However, our data show that the application of BI2536 almost completely abolishes the stiffness gradient, suggesting that it is rather unlikely that a cell density-independent stiffness gradient is responsible for the gradient in cell densities as suggested by the reviewer. However, the increase in cell density and thus tissue stiffness could provide a positive feedback signal, facilitating further cell proliferation as suggested by the reviewer. We have added this to the Discussion and provide a relevant reference.

We have not assessed whether changes in cell density correlate with changes in ECM deposition and if BI2536 impairs it. At the stages investigated in this study, there is very little ECM in the brain (in Koser et al., 2016, we added ECM to the tissue), and we would thus not expect a major effect either due to or impacting the ECM. However, we have added a potential contribution of the ECM to tissue stiffness in the Discussion.

2) In the last paragraph, you discuss applications of tiv-AFM to other possible contexts. Limitations of the system should be also mentioned.

This is a good point. We now discuss explicitly limitations of tiv-AFM, which are mainly that the method is restricted to surfaces and the time each map takes, which currently limits whole-brain scans at cellular spatial resolution to a temporal resolution of about half an hour.